# Binocular matching of thalamocortical and intracortical circuits in the mouse visual cortex

Yu Gu, Jianhua Cang*

Department of Neurobiology, Northwestern University, Evanston, United States

**Abstract** Visual cortical neurons are tuned to similar orientations through the two eyes. The binocularly-matched orientation preference is established during a critical period in early life, but the underlying circuit mechanisms remain unknown. Here, we optogenetically isolated the thalamocortical and intracortical excitatory inputs to individual layer 4 neurons and studied their binocular matching. In adult mice, the thalamic and cortical inputs representing the same eyes are similarly tuned and both are matched binocularly. In mice before the critical period, the thalamic input is already slightly matched, but the weak matching is not manifested due to random connections in the cortex, especially those serving the ipsilateral eye. Binocular matching is thus mediated by orientation-specific changes in intracortical connections and further improvement of thalamic matching. Together, our results suggest that the feed-forward thalamic input may play a key role in initiating and guiding the functional refinement of cortical circuits in critical period development.

*For correspondence: cang@northwestern.edu

**Competing interests:** The authors declare that no competing interests exist.

## Introduction

Two major transformations take place when visual information from the dorsal lateral geniculate nucleus (dLGN) of the thalamus reaches layer 4 of the primary visual cortex (V1). In one, V1 neurons become selective for bars and edges of certain orientations (*Ferster and Miller, 2000*; *Hubel and Wiesel, 1962*). In the other, eye-specific thalamic inputs converge onto individual V1 neurons, making them binocularly responsive (*Cang and Feldheim, 2013*). Combining the two transformations together, V1 neurons are now tuned to similar orientations through the two eyes (*Bridge and Cumming, 2001*; *Nelson et al., 1977*; *Wang et al., 2010*), a property presumably required for normal binocular perception. Importantly, binocularly-matched orientation tuning is not a necessary consequence of simply combining the two eyes' inputs. Instead, its establishment requires an active and experience-dependent process that takes place during a critical period in early life (*Wang et al., 2013*, *2010*). However, the circuit and synaptic mechanisms underlying the matching process are still unknown.

The mechanisms underlying *monocular* orientation selectivity in V1 have been extensively studied, providing a foundation for understanding its *binocular* matching. According to the 'feed-forward' model proposed by Hubel and Wiesel (*Hubel and Wiesel, 1962*), cortical orientation selectivity arises from the convergence of non-selective dLGN relay cells whose receptive fields are spatially aligned to generate larger peak excitation to the postsynaptic cell at its preferred orientation than at other orientations (*Priebe and Ferster, 2012*). This model is supported by a number of studies, especially in cats, including cross-correlation analysis of functional connectivity between dLGN and V1 neurons (*Alonso et al., 2001*; *Reid and Alonso, 1995*; *Tanaka, 1983*) and studies of visually evoked synaptic inputs in layer 4 cells during cortical inactivation (*Chung and Ferster, 1998*; *Ferster et al., 1996*). The feed-forward model appears to apply to mice as well, even though more

orientation-selective responses are seen in the mouse dLGN than in cats (*Piscopo et al., 2013*; *Scholl et al., 2013*; *Zhao et al., 2013a*). Two recent studies used optogenetic approaches to silence the visual cortex in order to isolate the thalamic excitation that could be recorded by in vivo whole-cell recording (*Li et al., 2013*; *Lien and Scanziani, 2013*). The thalamic input to layer 4 neurons was found to be tuned in a manner consistent with the feed-forward model, but not from orientation selective dLGN neurons (*Lien and Scanziani, 2013*). The tuned thalamic input is further amplified by similarly tuned intracortical input, thereby preserving its orientation tuning (*Li et al., 2013*; *Lien and Scanziani, 2013*).

Consequently, two different scenarios of thalamocortical transformation could underlie the binocularly matched orientation tuning in layer 4 neurons. In one, the above feed-forward model could hold true for both contralateral and ipsilateral pathways, where the thalamic and cortical inputs representing the same eyes are similarly tuned and both are binocularly matched. To achieve this scenario, experience-dependent changes for both thalamocortical and intracortical circuits must take place during the critical period to ensure their binocular matching. Alternatively, critical period plasticity may only drive synaptic changes of the intracortical circuits. This would mean that the two eye's thalamic inputs to layer 4 neurons may still be mismatched in their orientation tuning in adult mice, and the mismatch would be 'corrected' by cortical circuits. In other words, the feed-forward model would not be true for one or both eyes' thalamocortical transformation for binocular V1 neurons.

To test which of the two scenarios is true, we apply whole-cell recording and optogenetic approaches to binocular neurons in mouse V1. We find that both thalamic and cortical inputs are indeed binocularly matched in adult mice, thereby supporting the aforementioned first scenario. Surprisingly, we find that the thalamic input is already slightly matched at the beginning of the critical period, but this weak matching is not manifested due to mismatched cortical inputs. Binocular matching in the critical period is thus mediated by orientation-specific changes in these cortical connections and further improvement of thalamic matching.

## Results

We have previously discovered using extracellular recording that individual neurons in mouse V1 are initially tuned to different orientations through the two eyes until about postnatal day 21 (P21), and the difference declines progressively and reaches the adult level by P30 (*Wang et al., 2013*). To reveal the circuit mechanisms of this binocular orientation matching process, we have now performed in vivo whole-cell recording to examine the subthreshold membrane potential ($V_m$) responses and excitatory postsynaptic currents (EPSCs) underlying orientation tuning through the two eyes. We focused our experiments on the excitatory neurons in layer 4 because the synaptic mechanisms of their orientation tuning are better understood. We recorded at two developmental stages, P15-21 and P60-90, i.e., before and after the matching process, respectively.

### Binocular matching of the subthreshold membrane potential responses in layer 4 neurons

We first recorded visually evoked $V_m$ responses under current clamp (*Figure 1A–C*). In response to drifting gratings of different directions, layer 4 neurons in the binocular zone of visual cortex showed robust responses and orientation selectivity (*Figure 1D*). We then determined the orientation tuning of each cell's synaptic inputs, separately for the contralateral and ipsilateral eyes, by calculating the peak amplitude of cycle-averaged $V_m$ response (*Figure 1E*, see Materials and methods for details of analysis). As expected, the subthreshold $V_m$ were tuned to similar orientations through the two eyes in adult mice (*Figure 1E–F*). The inter-ocular difference of the preferred orientation, which we referred to as '*ΔO*', was smaller than 30° in the vast majority of the recorded cells (11/14 cells, *Figure 1F*), with a mean of 20.8 ± 3.3°, similar to that for the spiking responses in mice after the critical period (*Wang et al., 2013*).

On the other hand, in young mice prior to the onset of the critical period (P15-21), the orientation tuning of $V_m$ responses were completely mismatched binocularly. The *ΔO* distribution at this age group (*Figure 1G*, mean of 50.5 ± 3.8°, n = 25) was significantly skewed toward larger values compared to adults (p = 0.003; K-S test, *Figure 1H*), and it appeared to be largely random, which would

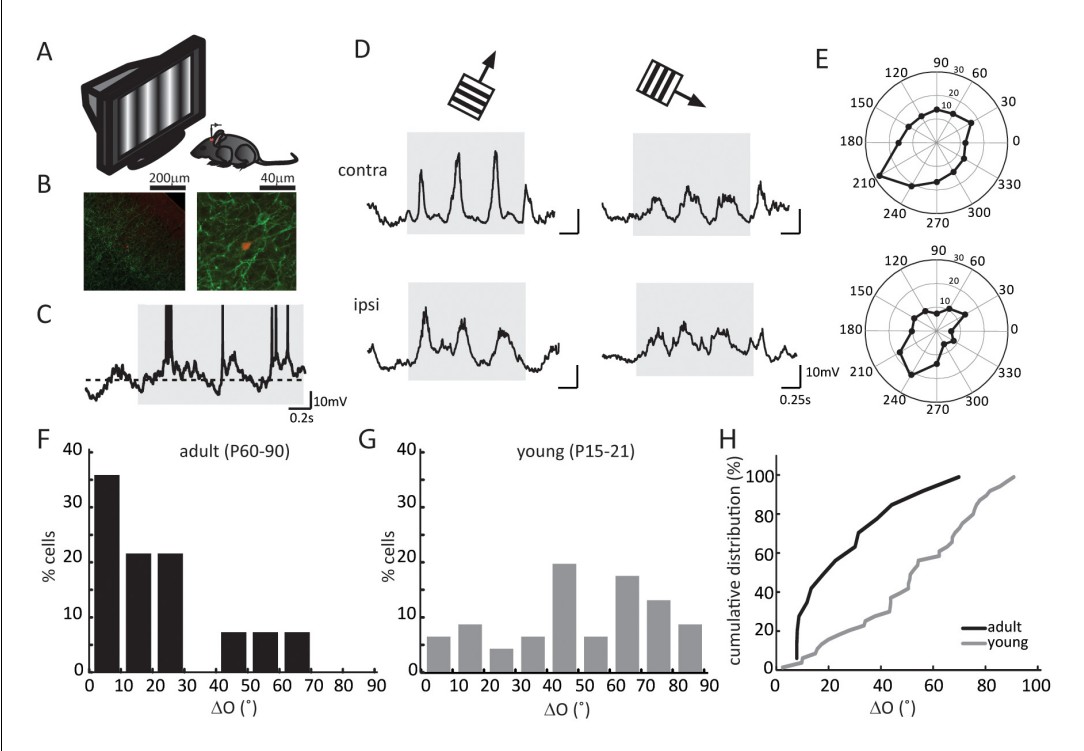

**Figure 1.** Binocular matching of subthreshold membrane potential responses in layer 4 neurons. (**A**) A schematic of the experimental setup. (**B**) Histology of a recorded neuron. Red is the neuron stained with Streptavidin-Alex Fluor 547; green is ChR2 conjugated with YFP. Left, low magnification showing the neuron's depth. Cortical surface is to the top right of the image. Right, high magnification showing the neuron's morphology. (**C**) An example of membrane potential ($V_m$) trace in response to drifting gratings. The gray shadow indicates the period of visual stimulation (same below in all figures). The dashed line is $-60$ mV. Action potential traces are truncated at $-10$ mV. (**D**) Subthreshold $V_m$ traces in response to the preferred (left) and orthogonal (right) directions. Top, stimulating contralateral eye only ('contra', same below); bottom, stimulating ipsilateral eye only ('ipsi'). (**E**) Polar plots of $V_m$ tuning curves. (**F–G**) Distribution of the inter-ocular difference of $V_m$'s preferred orientations ($\Delta O$) in adult (**F**) and young (**G**) mice. (**H**) Cumulative distribution of $\Delta O$ between adult (black, n = 14) and young (gray, n = 25) mice (p = 0.003, K-S test).

have been a uniform distribution between 0 and 90 with a mean of 45° (p = 0.11 compared to random distribution; K-S test, *Figure 1H*).

## Optogenetic isolation of thalamic inputs to layer 4 neurons in binocular V1

Layer 4 neurons receive several sources of synaptic inputs, including intracortical excitation from neighboring neurons and thalamocortical excitation from dLGN. As previously revealed in the monocular visual cortex, the orientation tuning of layer 4 cells is determined by the thalamic input, which is then linearly amplified by intracortical circuits (*Li et al., 2013*; *Lien and Scanziani, 2013*). To determine the thalamic contribution to binocular matching, we isolated layer 4 cells' thalamic input by voltage clamp recording coupled with optogenetic silencing. These experiments were performed in transgenic mice that expressed Channelrhodopsin-2 (ChR2) in neurons that express the GABA synthesizing enzyme glutamic acid decarboxylase GAD2. Consequently, by illuminating the exposed visual cortex with blue LED light, we were able to activate the inhibitory neurons and consequently silence the excitatory neurons, thereby allowing us to isolate thalamic excitation (*Li et al., 2013*; *Lien and Scanziani, 2013*).

Three control experiments were conducted to confirm the validity of this method. First, LED stimulation was able to completely silence the visually evoked responses of neurons recorded as deep as 700 μm below cortical surface, more than 200 μm deeper than layer 4 (*Figure 2A–B* and *Figure 2—figure supplement 1A–C*). Second, in addition to the inputs from dLGN and local excitatory neurons, layer 4 neurons could potentially receive excitatory inputs from the other hemisphere via

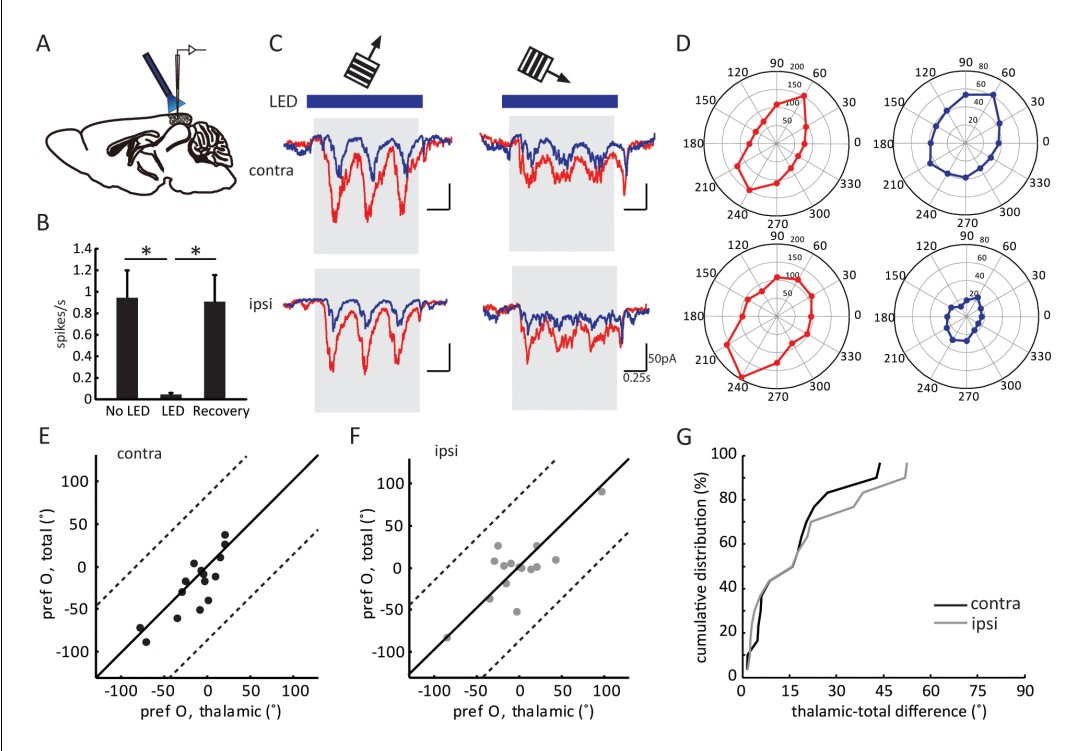

**Figure 2.** Optogenetic isolation of thalamic inputs to layer 4 neurons in binocular V1. (**A**) A schematic of optogenetic photostimulation and recording in V1. (**B**) Summary plot of average spike rate of deep-layer neurons before, during and after LED activation. The spike rate was calculated by averaging responses across all 12 directions of gratings, and then averaged for all the neurons (p = 0.0007, paired *t*-test between No LED and LED; and p = 0.0003, paired *t*-test between LED and recovery, n = 8). (**C**) Average EPSC traces in response to the preferred (left) and orthogonal (right) orientations of an example neuron. Top, stimulating contralateral eye; bottom, ipsilateral eye. Shadows indicate visual stimulation, blue bars indicate LED on. Red traces are responses without LED ('total EPSCs') and blue traces are responses with LED ('thalamic EPSCs'). (**D**) EPSC tuning curves for contralateral (top) and ipsilateral eye. Left, total EPSCs; right, thalamic EPSCs. (**E–F**) Correlation of orientation preference ('pref O', same below in similar figures) between thalamic and total EPSCs in adult mice. The dotted lines bound the region in which the data points can lie, i.e., 90° largest possible difference (same below in similar plots). (**E**) contralateral, (**F**) ipsilateral. (**G**) Cumulative distribution of orientation difference between thalamic and total EPSCs. No significant difference between contralateral (black) and ipsilateral (gray) responses (p = 0.89, K-S test, n = 15).

The following figure supplements are available for figure 2:

**Figure supplement 1.** Control experiments to validate optogenetic silencing of visual cortex.

**Figure supplement 2.** Scale factors for thalamo-cortical transformation in adult and young mice.

callosal projections, which would confound the interpretation of our results. We thus recorded layer 4 spiking activity while silencing the other hemisphere with LED photoactivation. Visually evoked responses were in fact unaffected by silencing the other hemisphere (*Figure 2—figure supplement 1D–F*), consistent with a previous report that callosal inputs are absent in layer 4 (*Mizuno et al., 2007*). Finally, to determine whether optogenetic activation of the inhibitory interneurons could influence thalamocortical transmission by potentially activating GABA$_B$ receptors on their axon terminals, we recorded visually evoked local field potentials (VEPs) in the visual cortex and examined the effect of administrating GABA$_B$ receptor antagonist CGP54624. In one set of experiments, we first applied the GABA$_B$ receptor agonist Baclofen to reduce the VEP amplitude, which was then reversed by the subsequent administration of CGP (*Figure 2—figure supplement 1G–H*), indicating the effectiveness of the antagonist (*Lien and Scanziani, 2013*). In another set of experiments, we found that optogenetic silencing of the visual cortex reduced the VEP by ~ 60 %, and importantly, it was not affected by the administration of CGP (*Figure 2—figure supplement 1I–J*). This result thus

indicates that thalamocortical transmission was not affected by optogenetic activation of the inhibitory neurons.

After confirming the completeness and validity of the optogenetic silencing approach, we performed in vivo whole-cell voltage clamp recording of layer 4 neurons in binocular V1 while presenting drifting gratings of different orientations. The recordings were done at the reversal potential of inhibition, which was determined by adjusting the holding potential to minimize the amplitude of the inhibitory postsynaptic currents evoked by LED photoactivation in the absence of visual stimulation (*Lien and Scanziani, 2013*). Consequently, the recorded currents were the total EPSCs (in the absence of LED photostimulation) or the thalamic EPSCs (in the presence of LED) received by individual layer 4 neurons.

As expected, silencing the visual cortex reduced the visually evoked EPSCs (e.g. *Figure 2C and D*). The degree of reduction, as measured by the ratio of mean thalamic EPSCs over mean total EPSCs during the visual stimulation across all directions (the 'scale factor'), was similar between the responses evoked through contralateral (0.32 ± 0.05) and ipsilateral eyes (0.32 ± 0.07; *Figure 2— figure supplement 2A–C*). Importantly, the thalamic EPSCs and total EPSCs were tuned to similar orientations in adult mice (*Figure 2E–F*), which is reflected by the small values of orientation preference difference between thalamic and total EPSCs. Furthermore, this difference was similar between contralateral (15.2 ± 3.4°) or ipsilateral eye (17.7 ± 4.5°) responses (*Figure 2G*, p = 0.89, K-S test). The correlation between thalamic and total EPSCs was consistent with the previous results obtained in the monocular visual cortex, and we found here that it held true for binocular neurons through both eyes in adult mice.

## Binocular matching of thalamic input precedes that of intracortical input

We next analyzed the binocular relationship of both thalamic and total EPSCs in adult mice. As expected from the matched spiking and $V_m$ responses, the total EPSCs of layer 4 cells were tuned to similar orientations through the two eyes (*Figure 3A1*), with its inter-ocular difference (i.e. $\Delta$O) of 20.6 ± 4.3° (n = 15, *Figure 3A2*). The thalamic EPSCs were also tuned to similar orientations through the two eyes (*Figure 3B1–B2*, $\Delta$O = 19.6 ± 5.5°, n = 15), consistent with the fact that thalamic excitation determines cortical tuning for both contralateral and ipsilateral responses. Indeed, the $\Delta$O of thalamic EPSCs were significantly correlated with that of total EPSCs for individual neurons (*Figure 3E*, r = 0.57, p = 0.025). In other words, the thalamic inputs to layer 4 neurons are tuned to similar orientations through the two eyes in adult mice, leading to binocularly-matched orientation preference in these cells.

The same whole-cell recording and optogenetic silencing were then performed in mice between P15 and P21, before the binocular matching of spiking and $V_m$ responses. First, we found that the ratio of thalamic over total EPSCs, i.e., the scale factor, was slightly higher and less matched binocularly in the young mice (0.43 ± 0.05 for the contralateral pathway and 0.52 ± 0.08 for ipsilateral, *Figure 2—figure supplement 2D–F*), suggesting the immaturity of thalamocortical transformation at this age. Second, we analyzed the correlation between the orientation preference of total EPSCs and that of the thalamic EPSCs, separately for contralateral and ipsilateral eyes (*Figure 3—figure supplement 1A–B*). For both eyes, many more data points deviated from the unity line (i.e., identical tuning between the two) in young mice than in adults (compared with *Figure 2E and F*). To quantify this, we calculated the difference between thalamic and total EPSCs tuning for individual cells and found that it was significantly larger in the young mice than in the adult, and for both eyes (*Figure 3—figure supplement 1C*, p = 0.040 for contralateral eye and p = 0.043 for ipsilateral eye, K-S test).

Next, we examined the binocular relationship of total and thalamic EPSCs in the young mice. The total EPSCs were not yet matched in their orientation preference at this age (*Figure 3C1*), with their inter-ocular difference ($\Delta$O) quite randomly distributed within the entire range between 0 and 90 (*Figure 3C2*, $\Delta$O = 41.4 ± 5.5°, n = 26, p = 0.023 compared to adult, p = 0.45 compared to random, K-S test). For the thalamic EPSCs, their orientation tuning was not as well matched as in adult mice either (*Figure 3D1–D2*, $\Delta$O = 30.9 ± 4.0°, n = 26, p = 0.037 compared to adult, K-S test). However, their $\Delta$O distribution was skewed toward 0 (*Figure 3D2*), indicating a certain degree of thalamic matching already present in these mice (p = 0.003 compared to random, K-S test). To better illustrate this, we plotted the $\Delta$O of total EPSCs against that of thalamic $\Delta$O for individual cells (*Figure 3F*, r = 0.04, p = 0.84, i.e., no correlation between them). There were more cells above the

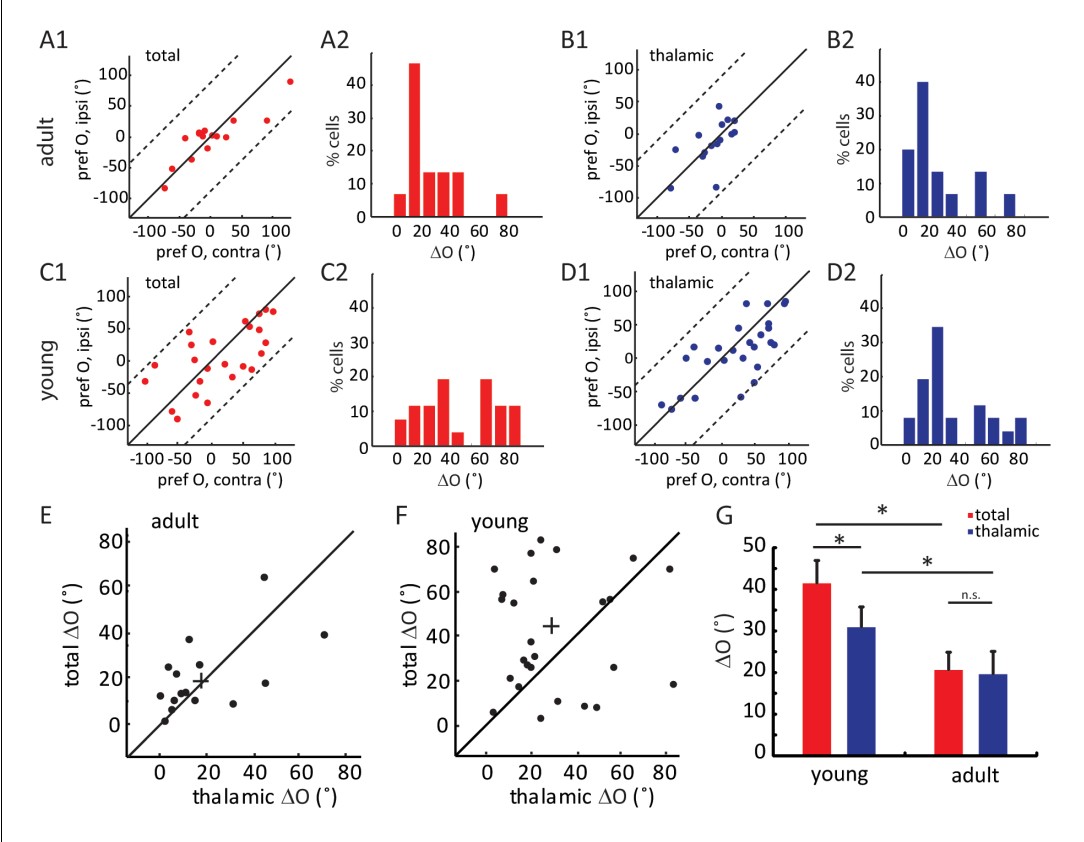

**Figure 3.** Binocular matching of thalamic input precedes that of intracortical input. (**A–B**) Binocular matching of total (A1 and A2, red) and thalamic (B1 and B2, blue) EPSCs in adult mice. (**A1**) and (**B1**) show correlations of the preferred orientation between the two eyes. (**A2**) and (**B2**) are the distributions of the inter-ocular difference (ΔO). (**C–D**) Same plots as (**A–B**), but in young mice between P15-P21. (**E–F**) Correlation of 'thalamic ΔO' and 'total ΔO' of individual cells in adult (**E**) and young mice (**F**). Cross indicates means of the data for the two axes. (**G**) Mean ΔO of the total (red) and thalamic (blue) EPSCs from young and adult mice. Significant differences are seen between the two age groups and between the total and thalamic ΔO in the young mice. p = 0.038, paired *t*-test between total and thalamic ΔO in young mice, n = 26; p = 0.83, paired *t*-test between total and thalamic ΔO in adult mice, n = 15; and p = 0.023, K-S test between young and adult mice for total ΔO; p = 0.037, K-S test between young and adult mice for thalamic ΔO. Error bars are S.E.M.

The following figure supplements are available for figure 3:

**Figure supplement 1.** Orientation difference between thalamic and total EPSCs in young mice.

**Figure supplement 2.** Relative amplitude of thalamic and total EPSCs between the two eyes over development.

unity line, i.e., smaller ΔO values for thalamic EPSCs, indicating that the thalamic EPSCs were better matched than total EPSCs at this age (p = 0.038, paired *t*-test, *Figure 3G*). Finally, when analyzing the relative strength of the two eyes' inputs (i.e. ocular dominance), we found that the contralateral EPSCs were about twice stronger than the ipsilateral ones for both thalamic and total inputs, and this ratio remained consistent at both ages (*Figure 3—figure supplement 2*). In other words, the ocular dominance of thalamic and total EPSCs does not undergo any gross change during development when they become binocularly matched.

In adult mice, intracortical excitation to layer 4 cells is tuned to the same orientation as the thalamic excitation, thereby maintaining the orientation tuning of the thalamic input (*Li et al., 2013*; *Lien and Scanziani, 2013*). The lack of adult-level 'thalamic-total' correlation in the young mice suggests that the intracortical circuits are not yet organized in the same manner. To examine the tuning of intracortical excitatory input across development, we did a point-by-point subtraction between thalamic and total EPSCs (*Figure 4—figure supplement 1A*). This was done for all 12 stimulus

directions to determine the tuning curves and orientation preference of intracortical EPSCs (*Figure 4—figure supplement 1B*). As expected, the intracortical EPSCs were tuned to similar orientations as the thalamic EPSCs in adults, for both contralateral and ipsilateral eyes (*Figure 4A–C*, 'thalamic-cortical difference': 17.0 ± 4.4° for contralateral pathway and 22.4 ± 6.0° for ipsilateral), thus binocularly matched (*Figure 4—figure supplement 1C*). In contrast, in the young mice, the correlation between thalamic and cortical orientation preferences was weak for both eyes (*Figure 4D–E*, 35.8 ± 5.5° for contralateral and 49.0 ± 5.3° for ipsilateral). Interestingly, for individual cells, the difference

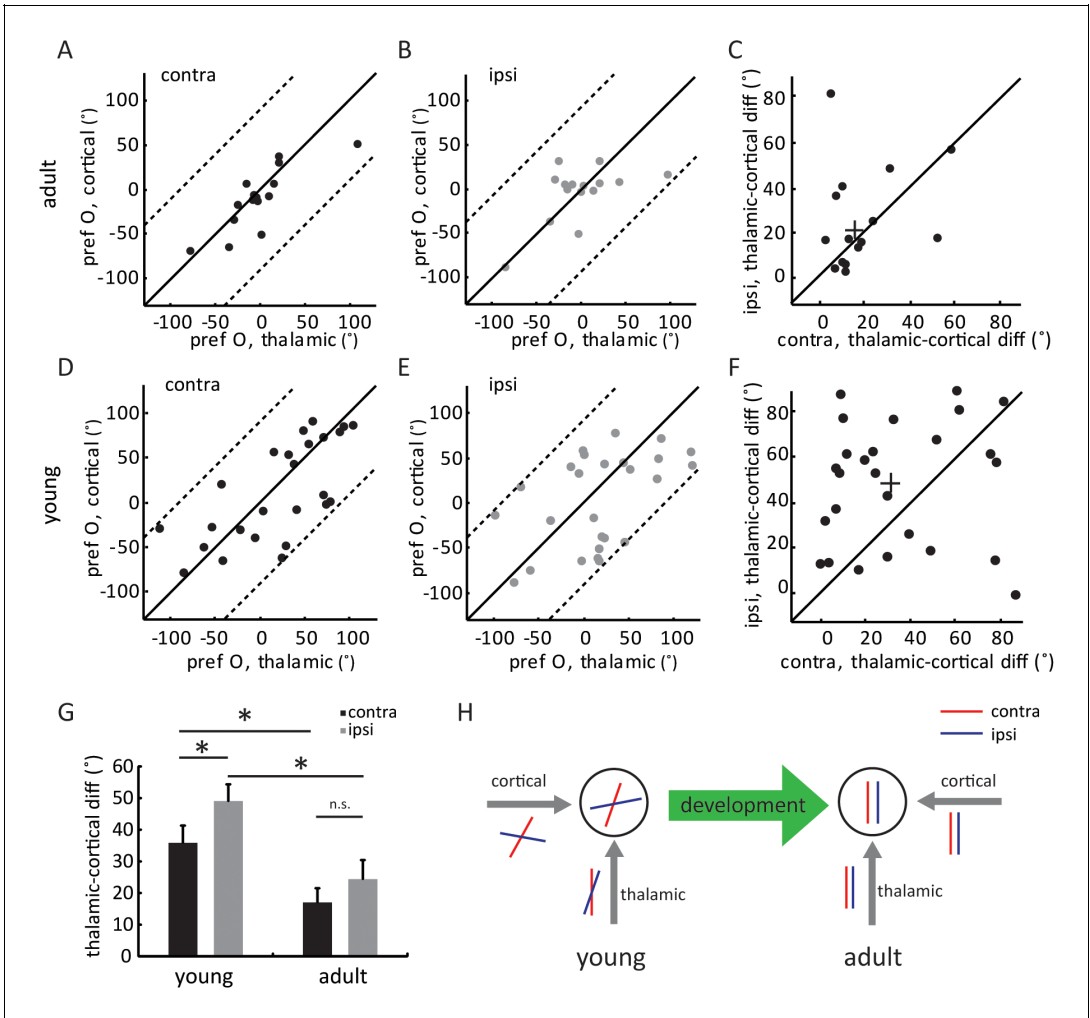

**Figure 4.** Development of eye-specific thalamocortical transformation. (**A–B**) Correlation of orientation preference between thalamic and cortical EPSCs in adult mice. (**A**) for contralateral and (**B**) for ipsilateral pathway. (**C**) Scatter plots of thalamic-cortical difference in orientation preference for the two eyes in adult mice. The cross indicates the mean of the distribution along the two axes. (**D–F**) Same plots as (**A–C**), but in young mice between P15 and P21. (**G**) Mean thalamic-cortical difference in orientation preference from each eye in young and adult mice. Significant differences are seen between the two age groups and between the contralateral (black) and ipsilateral (gray) eyes in the young mice. p = 0.041, paired *t*-test between the eyes in young mice, n = 26; p = 0.29, paired *t*-test between the eyes in adult mice, n = 15; and p = 0.043, K-S test between young and adult for contralateral eye; p = 0.016, K-S test between young and adult for ipsilateral eye. Error bars are S.E.M. (**H**) A summary schematic of thalamic and cortical binocular matching and thalamo-cortical transformation over development. In mice before the critical period (left), the orientation tuning of the two eyes' thalamic inputs to layer 4 neurons are already slightly matched (represented by the red and blue oriented bars). The slight matching is not manifested in the cell's tuning due to mismatched intracortical inputs. Further improvement of thalamic matching and rewiring of the intracortical circuits to follow the thalamic tuning give rise to binocularly matched tuning in adult mice (right).

The following figure supplement is available for figure 4:

**Figure supplement 1.** Binocular matching of intracortical EPSCs in layer 4 neurons.

between thalamic and intracortical tuning was greater for the ipsilateral eye than for the contralateral eye (*Figure 4F–G*, p = 0.041, paired *t*-test). As a result, the intracortical input was completely mismatched binocularly in the young mice (*Figure 4—figure supplement 1D*), despite the weak matching of thalamic input. The mismatched intracortical input thus leads to a mismatch in its total excitation and the resulting $V_m$ and spike responses in mice before the critical period.

## Discussion

In this study, we combined optogenetic silencing and whole-cell recording to isolate thalamic and cortical excitatory inputs to individual layer 4 neurons and examined their binocular matching during development. Our results reveal two developmental profiles of these circuits. First, the binocular matching of thalamic inputs initiates before that of intracortical circuits. Second, the intracortical circuits serving the contralateral eye mature before those for the ipsilateral eye. In mice before the critical period, the slight binocular matching of thalamic input is not manifested due to random connections in the cortex, especially those serving the ipsilateral eye (*Figure 4H*). In adult mice, the thalamic and cortical inputs serving the same eyes are tuned to similar orientations and are both matched binocularly (*Figure 4H*), thereby giving rise to normal binocularity. Binocular matching is therefore mediated by further improvement of thalamic matching and orientation-specific changes of cortical circuits during the critical period.

### Early binocular matching of thalamic inputs

Our results demonstrate that the thalamic inputs to layer 4 neurons initiate their binocular matching before intracortical circuits. This finding adds to the growing literature that the visual system develops in a hierarchical manner where neural circuits in earlier stages of visual pathway mature sooner than those in later stages (*Daw, 1997*; *Lewis and Maurer, 2005*). For example, we previously revealed a sequence of binocular matching during the critical period (*Wang et al., 2013*): simple cells match before complex cells, with the two classes of cells representing successive stages of visual processing in V1. Here, our new data indicate that this sequence can be extended to the 'pre-critical period' for the matching of thalamocortical projections, and very importantly, at the level of individual cells. Similarly, a developmental maturation sequence of synaptic plasticity has been shown by in vitro studies following the visual pathway, from LGN to layer 4 and then from layer 4 to layer 2/3 (*Jiang et al., 2007*; *Kirkwood and Bear, 1994*), which may underlie some aspects of the functional maturation we observed in vivo.

What mechanisms could give rise to the early binocular matching of the thalamic inputs? It is possible that the visual experience between eye opening (P12-14) and our recording time (P15-21) may contribute to this early thalamic binocular correspondence. Alternatively, previous studies have shown that many aspects of visual system organization and function can be established by experience-independent mechanisms, such as ocular dominance columns in cat visual cortex (*Crowley and Katz, 2000*), monocular orientation selectivity (*Ko et al., 2014*; *Wang et al., 2010*) and receptive field structures (*Sarnaik et al., 2014*) in mouse V1. The experience-independent mechanisms could include molecular guidance cues and activity-dependent processes driven by patterned spontaneous activity such as retinal waves (*Cang and Feldheim, 2013*). Structured spontaneous activities have been observed in the dLGN and visual cortex before vision onset (*Ackman et al., 2012*; *Chiu and Weliky, 2001*; *Hanganu et al., 2006*; *Siegel et al., 2012*; *Weliky and Katz, 1999*). A correlation-based computational study supports the notion that patterned activity could give rise to monocular receptive fields that are consistent with the feed-forward model (*Miller, 1994*). However, an activity-driven binocular matching of thalamic inputs requires some correlation between the two eye-specific pathways. Amazingly, certain degrees of binocular correlation may indeed exist in mice even at the level of retinal waves (*Ackman et al., 2012*), possibly via direct retino-retinal connections (*Müller and Holländer, 1988*), retinopetal modulations (*Gastinger et al., 2006*), or light-driven activity in the intrinsic photosensitive retinal ganglion cells through the closed eye lids (*Delwig et al., 2012*; *Renna et al., 2011*). Future studies are needed to investigate whether such spontaneous activity is involved in the early binocular matching of thalamic inputs to cortical neurons.

## Earlier maturation of contralateral intracortical circuits

In adult mice, the intracortical excitation that individual layer 4 neurons receive prefers the same orientation as the thalamic input ([*Li et al., 2013*; *Lien and Scanziani, 2013*], and *Figure 2*), thereby preserving its orientation tuning. In other words, layer 4 excitatory neurons must be preferentially connected with cortical neurons that prefer similar orientations. Here, we found that the correlation between thalamic and cortical tuning, and presumably the underlying neural circuits, have not been established in mice before the critical period, indicating that the classic 'Hubel and Wiesel' feed-forward model do not exist in animals right after eye opening. These results are consistent with the connectivity patterns of mouse layer 2/3 neurons and their developmental profile. The preferential connectivity among similarly tuned layer 2/3 neurons seen in adult mice (*Ko et al., 2011*; *Lee et al., 2016*) does not exist right after eye opening (*Ko et al., 2013*), and it requires visual experience to establish (*Ko et al., 2014*). Our results thus indicate that the experience-dependent rewiring of intracortical circuits, together with the further improved matching of thalamic inputs, underlie binocular matching of orientation tuning in layer 4 neurons.

Interestingly, we found that the correlation between thalamic and cortical tuning is slightly better for the contralateral pathway in the young mice, indicating that the contralateral pathway starts to mature earlier than the ipsilateral one. A similar developmental sequence of the two pathways has been revealed in both cat and mouse visual cortices. In mouse ocular dominance plasticity, while both contralateral and ipsilateral circuits respond to imbalanced visual inputs in juveniles (*Frenkel and Bear, 2004*), only the ipsilateral circuits remain responsive in young adults (*Sawtell et al., 2003*), suggesting that the ipsilateral pathway reaches maturity later. In cats, visual cortical maps at vision onset are dominated by the contralateral eye and visual experience is needed for the other eye's response to strengthen (*Crair et al., 1998*). Given the strong contralateral dominance, it is possible that the contralateral pathway may be used as a 'guide' for ipsilateral development. Consistent with this idea, it has been shown that the refinement of the ipsilateral retinotopic map in the mouse visual cortex can be delayed or accelerated by manipulating the contralateral input (*Smith and Trachtenberg, 2007*). It is conceivable that a similar interaction between the two pathways exists in binocular matching, where the dominant input may guide the weaker one in rewiring the thalamocortical and intracortical circuits. Future experiments where the matching process can be followed chronically will be needed to test this hypothesis.

Finally, it is important to note that, even though it has been widely used in neuroscience research, voltage clamp is normally incomplete in whole-cell recordings due to the so-called 'space clamp' issue (*Williams and Mitchell, 2008*). This problem is especially severe for dendritic neurons and for in vivo recording, where the series resistance is often higher than in slices. As a result, our measurement likely underestimated the amplitude of EPSCs that layer 4 neurons receive from thalamic and intracortical inputs. Furthermore, the estimation of intracortical EPSCs by linear subtraction could be confounded by potential non-linear interactions between thalamic and cortical inputs. In other words, the exact EPSC amplitudes and their scale factor and ocular dominance ratio may not be accurately reflected in our study. On the other hand, the calculation of preferred directions should not be much affected by this issue, because the EPSC underestimation was likely similar across different directions. Therefore, the main discoveries in our study, i.e., the two developmental profiles which are entirely based on the preferred directions, should hold true despite the limitations of voltage clamp recordings.

In conclusion, we have discovered the thalamocortical and intracortical circuits underlying binocular matching and revealed their developmental profiles. Our results suggest that the feed-forward thalamocortical pathway may play an important role in the development of visual circuits, possibly through initiating and guiding the functional rewiring and refinement of cortical circuits in visual cortex. These findings provide an exciting foundation for future mechanistic studies of critical period plasticity and binocular matching.

## Materials and methods

### Mice and preparations

Wild type C57BL/6 and 'GAD2-ChR2' double heterozygous mice of different age groups (P15-21, P60-90) and both sexes were used in the experiments. The double heterozygous were obtained by

crossing *Gad2^tm2(cre)Zjh* and *Gt(ROSA)26Sor^tm32(CAG-COP4\*H134R/EYFP)Hze* homozygous mice originally from the Jackson Laboratory. Animals were raised on a 12 hr light/dark cycle, with food and water available ad libitum. All animals were used in accordance with protocols approved by Northwestern University Institutional Animal Care and Use Committee (IS00003509).

For recordings, mice were sedated with chlorprothixene (5 mg/kg in water, i.m.) and then anesthetized with urethane (1–1.25 g/kg in 10 % saline solution, i.p.). Atropine (0.3 mg/kg, in 10 % saline) and dexamethasone (2 mg/kg, in 10 % saline) were administrated subcutaneously, as described before (*Wang et al., 2013*, *2010*). The scalp was shaved and skin removed to expose the skull. A metal head plate was implanted on top of the skull with Metabond (Parkell, Edgewood, NY), and the plate was then mounted to a stand on the vibration isolation table. A thin layer of silicone oil was applied on both eyes to prevent from drying. A small craniotomy (~1.5 mm$^2$) was made on the left hemisphere to expose binocular V1. The center of the craniotomy was 3.0 mm lateral and 0.5 mm anterior from the Lambda point. Throughout recordings, toe-pinch reflex was monitored and additional urethane (0.2–0.3 g/kg) was supplemented as needed. The animal's temperature was monitored with a rectal thermoprobe and maintained at 37°C through a feedback heater (Frederick Haer Company, Bowdoinham, ME).

## Whole-cell recording

Blind whole-cell patch clamp was performed to record cortical cells intracellularly as described previously (*Zhao et al., 2013b*). Glass pipettes had tip openings of 1.5–2 μm (4–8 MΩ). The internal solution contained 135 mM K-gluconate, 4 mM KCl, 0.5 mM EGTA, 10 mM HEPES, 10 mM Na-phosphocreatine, 4 mM Mg-ATP and 0.4 mM GTP. The pH was adjusted to 7.2 with KOH, and the osmolarity was adjusted to 290–300 mOsm/L. After inserting the pipette perpendicularly to the horizontal plane of the mouse head into the cortex, 2.0 % agarose in artificial cerebrospinal fluid (ACSF, containing 140 mM NaCl, 2.5 mM KCl, 11 mM Glucose, 20 mM HEPES, 2.5 mM CaCl$_2$, 3 mM MgSO$_4$, 1 mM NaH$_2$PO$_4$) was applied on top of the cortex to reduce pulsation. Signals were amplified using MultiClamp 700B (Molecular Devices, Sunnyvale, CA), sampled at 10 kHz, and then acquired with System three workstation (Tucker Davis Technologies, Alachua, FL). Pipette capacitance and the open tip resistance were compensated initially. After the whole-cell configuration was achieved, the membrane potential was recorded under current-clamp mode with no current injected. For recording EPSCs, the cell was voltage clamped at ~ −63 mV (average reverse potential for inhibition, adjusted individually by minimize the amplitude of the inhibitory postsynaptic current evoked by photostimulation of GAD2+ neurons, consistent with (*Li et al., 2013*; *Lien and Scanziani, 2013*). Only cells with stable responses (< 15 % change in baseline) and low series resistant (< 50 MΩ for voltage clamp, and < 100 MΩ for current clamp) were included in our analysis. Note that the reported values were not corrected for the junction potential. The depths of recorded cells were between 350 and 500 μm (reading from the micromanipulator) from the point where the pipette broke into the pia surface. The morphology of biocytin-stained cells further confirmed the accuracy of the manipulator readings and all stained cells were in layer 4.

## Extracellular recording

Tungsten electrodes (5–10 MΩ, FHC, Bowdoinham, ME) were inserted perpendicular to the pial surface. For single unit recordings to confirm complete silencing, signals were recorded between 600 μm and 700 μm in depth, corresponding to deep layers (layer 5 and 6). For single unit recordings to study the impact of callosal projections and for field recordings of visually evoked potentials (VEPs), signals were recorded between 350 μm and 450 μm in depth, corresponding to layer 4. Electrical signals were filtered between 0.3 and 5 kHz for spikes, and 10 and 300 Hz for VEPs and sampled at 25 kHz using a System three workstation (Tucker Davis Technologies, FL). The spike waveforms were sorted offline in OpenSorter (Tucker Davis Technologies, Alachua, FL) to isolate single units as described before (*Wang et al., 2013*, *2010*).

## Visual stimuli

Sinusoidal gratings drifting perpendicular to their orientations were generated with Matlab Psychophysics toolbox (*Brainard, 1997*), as described previously (*Wang et al., 2010*). Stimuli were presented using a CRT monitor (40 × 30 cm, 60 Hz refresh rate, ~35 cd/m$^2$ luminance) placed 25 cm in

front of the animal. The direction of the gratings varied between 0° and 330° (12 steps at 30° spacing) in a pseudorandom sequence. Spatial frequency of the stimuli was 0.02 cycle/degree. Temporal frequency was fixed at two cycle/s. Each stimulus was presented for 1.5 s (three cycles), with 1.5 s inter-stimulus interval or 4.5 s interval when LED was used in order to allow the optogenetically activated inhibitory cells to recover.

## Photostimulation

To photostimulate ChR2-expressing cells, an optic fiber (0.2 mm core diameter) driven by a blue LED (470 nm, Doric Lenses, Quebec, Canada) was placed ~0.5 mm above the exposed cortex. The tip of the LED fiber was placed at a similar position in all mice. During recordings, it was buried in the agarose that was applied to reduce the pulsation of the brain and protect the tissue. To prevent direct photostimulation of eyes by the LED light, Metabond used for mounting the head plate was prepared with black ink, and a piece of thick black paper was carefully placed around the fiber to ensure no light could be seen from the front and lateral sides, as described before (*Zhao et al., 2014*). The LED was driven by a series of square waves (50 Hz) starting from 100 ms before the onset of each visual stimulus and ending at 100 ms after the offset of each visual stimulus. The intensity of LED light was ~160 mW/mm$^2$ at the tip of the optic fiber in all recordings, which was confirmed to be reliably effective in silencing all excitatory neurons in visual cortex (*Figure 2* and *Figure 2—figure supplement 1*).

## Histology

2.0% biocytin was added in the internal solution to label recorded cells. After in vivo whole-cell recordings, mice were overdosed with euthanasia solution (150 mg/kg pentobarbital) and perfused with PBS and then 4 % paraformaldehyde (PFA) solution. The brain was fixed in 4 % PFA overnight. Coronal slices 150 µm thick were cut from the fixed brain using a vibratome (VT1000S, Leica Microsystems, Wetzlar, Germany). The labeled cells were revealed by visualizing biocytin with Streptavidin-Alex Fluor 547 conjugate (Invitrogen, Carlsbad, CA). Fluorescence images were captured using a Zeiss LSM5 Pascal confocal microscope (Carl Zeiss, Jena, Germany) in z-series scanning.

## Data analysis

Whole-cell recording data were first analyzed using a custom MATLAB program (originally written by a former lab member Dr. Xinyu Zhao). For current-clamp data, spikes were detected by calculating the first derivative of raw voltage traces (dV/dt), and the start of a spike was the time point when dV/dt reached a manually set positive threshold. Subthreshold $V_m$ were extracted by removing spikes from the raw voltage traces by a 6 ms median filter. With each stimulus trial included three cycles of drifting gratings (1.5 s stimulus duration and 2 Hz temporal frequency), the subthreshold $V_m$ traces were cycle-averaged for each stimulus condition (from 50 to 550 ms after each stimulus cycle onset). Note that only the first two cycles were used to cycle-average the responses if the cell showed adaptation in the third cycle (*Lien and Scanziani, 2013*). The averaged $V_m$ trace for the blank stimulus (i.e. gray screen) was used to calculate the mean ($V_m$ baseline) and standard deviation of spontaneous $V_m$ fluctuations. The $V_m$ baseline was then subtracted from the smoothed and cycle-averaged $V_m$ trace for each visual stimulus condition, i.e., gratings of certain direction, and the peak amplitude of the resulting $V_m$ trace was used as the response magnitude for that direction.

For voltage-clamp data, the current traces were firstly smoothed by a 40 ms mean filter (*Li et al., 2013*, *2015*), and then cycle-averaged for each stimulus condition as described above. The $I_m$ baseline was calculated as the mean of the $I_m$ trace to the blank stimulus, separately for conditions with LED photoactivation ('LED-on'), and then subtracted from the cycle-averaged trace of each condition to obtain visually evoked EPSC responses. The peak EPSC amplitude was used for plotting tuning curves and subsequent analysis. Finally, the intracortical EPSC traces were generated by a point-by-point subtraction of thalamic EPSCs (LED-on) from the total EPSCs (LED-off) traces (*Figure 4*). The scale factors were calculated as the mean thalamic EPSC amplitude of the tuning curve divided by mean total EPSC amplitude of the tuning curve for each recorded neuron.

To determine the preferred orientation and degree of selectivity, we calculated $\frac{\Sigma R(\theta)e^{2*i*\theta}}{\Sigma R(\theta)}$, where $R$ ($\theta$) is the response magnitude of $V_m$ or EPSC, at $\theta$ direction of gratings. Its amplitude was used as a global orientation selective index (gOSI). Half of its complex phase was calculated (*Niell and*

*Stryker, 2008*) and then converted to the preferred orientation (pref_O) by subtracting 90°, to confine pref_O between −90° to 90°. The difference in the preferred orientation between the two eyes was calculated by subtracting ipsilateral pref_O from contralateral pref_O along the 180° cycle (−90° to 90°). The absolute values of these differences (ΔO) were used in all quantifications.

No statistical methods were used to predetermine sample sizes, but our sample sizes are similar to those reported in the field (*Li et al., 2013*; *Lien and Scanziani, 2013*). All values were presented as mean ± SEM. Differences between groups were tested for significance using the Kolmogorov-Smirnov test (K-S test) and paired *t*-test, as stated in the text. Statistical analyses and graphs were made with Matlab (Mathworks, Natick, MA).

## Acknowledgements

We thank Xinyu Zhao for help with data analysis and Xuefeng Shi for help with figures. We thank Tom Bozza for the use of confocal microscope. We gratefully acknowledge Dr. Li Zhang and Dr. Huizhong Tao from University of Southern California for their help with in vivo whole-cell voltage clamp recording. This research was supported by US National Institutes of Health (NIH) grants (EY020950 and EY026286 to JC).

## Additional information

### Funding

| Funder | Grant reference number | Author |
| --- | --- | --- |
| National Institutes of Health | EY020950 | Jianhua Cang |
| National Institutes of Health | EY026286 | Jianhua Cang |

The funders had no role in study design, data collection and interpretation, or the decision to submit the work for publication.

### Author contributions

YG, Conceptualization, Data curation, Formal analysis, Investigation, Visualization, Methodology, Writing—original draft; JC, Conceptualization, Data curation, Formal analysis, Supervision, Funding acquisition, Writing—review and editing

### Author ORCIDs

Jianhua Cang, http://orcid.org/0000-0002-0760-7468

### Ethics

Animal experimentation: All animals were used in accordance with protocols approved by Northwestern University Institutional Animal Care and Use Committee (IS00003509).

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
