## [Decision Letter]

Thank you for submitting your article "Binocular Matching of Thalamocortical and Intracortical Circuits in the Mouse Visual Cortex" for consideration by *eLife*. Your article has been reviewed by three peer reviewers, and the evaluation has been overseen by a Reviewing Editor and Eve Marder as the Senior Editor. The following individuals involved in review of your submission have agreed to reveal their identity: Sacha B Nelson (Reviewing Editor and Reviewer #1) and Huizhong Tao (Reviewer #3).

The reviewers have discussed the reviews with one another and the Reviewing Editor has drafted this decision to help you prepare a revised submission.

Summary:

Visual cortical neurons are selective for orientation and respond to stimulation through both eyes. The developmental process by which inputs through the two eyes become matched in their orientation tuning is incompletely understood. Here, the authors use optogenetics and in vivo whole cell recordings to show that this matching occurs at earlier ages for thalamic input than for cortical input.

Essential revisions:

1) The authors use CGP 55845 to block GABA-B receptors and find no effect on field potentials when intracortical inputs are silenced via the LED. An important positive control for the effectiveness of the antagonist would be to show that it does have an effect when the LED is not turned on. This, or some other control demonstrating the drug was effective should be included.

2) The text should be revised so that the order in which the results are discussed is the same for the older and younger animals. The organization of the figures is fine but the order in the text was confusing.

The EPSC properties reported for adult mice are presented in the following order:

1) The ipsilateral and contralateral scaling factor

2) Individual eye matching of the thalamic and total EPSCs (both ipsi and contra)

3) Inter-ocular difference of the total EPSC

4) Inter-ocular difference of the thalamic EPSC

5) Correlation of inter-ocular difference between total and thalamic EPSC

These properties should be presented in the exact same order also for young mice.

Furthermore, the intracortical EPSC and its individual eye matching with the thalamic EPSC as well as the inter-ocular difference of the intracortical EPSC should be presented with the rest of the EPSC properties in the adult first and then in the young animal.

3) These are difficult experiments and the authors should acknowledge somewhere the limitations of their approach, ideally in the Discussion.

A) Voltage clamp in vivo may be incomplete due to space clamp issues. This could even have larger effects on intracortical than thalamic input since the latter are likely to occur more proximally.

B) The authors should also acknowledge the limitation of linear subtraction as a method for estimating the cortical contribution. The interaction between thalamic and cortical input is likely not linear.

C) The discussion of possible early matching mechanisms in the Discussion is fine, but the statement "Amazingly, certain degrees of binocular correlation may indeed exist in mice even at the level of retinal waves (Ackman et al., 2012), possibly via direct retino-retinal connections (Muller and Hollander, 1988), retinopetal modulations (Gastinger et al., 2006), or light-driven activity in the intrinsic photosensitive retinal ganglion cells through the closed eye lids (Delwig et al., 2012; Renna et al., 2011)" implies that all such correlation must have occurred before eye opening. However, the experience between eye opening and recording may also contribute.

D) Finally, the following conclusion is not warranted, although it could be stated as a speculation: "feed-forward thalamocortical pathway plays an important role in initiating and guiding the functional rewiring and refinement of cortical circuits in visual system development." Correlation does not equal causation.

4) The scale factor was calculated as the maximal thalamic EPSC amplitude of the tuning curve divided by maximal total EPSC amplitude. Since preferred orientation often differed between thalamic EPSC and total EPSC, especially for young cells, the scale factor was often calculated from two responses evoked by different stimuli. To have a more accurate measurement of scale factor, the authors can select 4 o 5 best responses of the tuning curves for thalamic EPSC and total EPSC (evoked by the same stimuli), do some averaging and then calculate the ratio.

---

## [Author Response]

*Essential revisions:*

*1) The authors use CGP 55845 to block GABA-B receptors and find no effect on field potentials when intracortical inputs are silenced via the LED. An important positive control for the effectiveness of the antagonist would be to show that it does have an effect when the LED is not turned on. This, or some other control demonstrating the drug was effective should be included.*

We agree with the reviewers that this is an important positive control. When we analyzed CGP’s effect on VEP amplitude in the absence of LED, we found that it was not different between control and CGP conditions. This is in fact consistent with the literature that administration of GABA-B receptor antagonist alone does not affect the field potential amplitudes (Liu et al., 2007; Yamouchi et al., 2000). In other words, GABA-B receptors are not normally active during our recording condition. We have therefore performed new experiments to demonstrate the effectiveness of the antagonist, following Lien and Scanziani (2013). In this set of experiments, we first applied the GABA-B receptor agonist Baclofen, which reduced the VEP amplitude. Subsequent administration of CGP was able to reverse Baclofen’s effect, thus indicating the effectiveness of the antagonist. These new data are added Figure 2—figure supplement 1 and included in the text (subsection “Optogenetic isolation of thalamic inputs to layer 4 neurons in binocular V1”, second paragraph).

*2) The text should be revised so that the order in which the results are discussed is the same for the older and younger animals. The organization of the figures is fine but the order in the text was confusing.*

*The EPSC properties reported for adult mice are presented in the following order:*

*1) The ipsilateral and contralateral scaling factor*

*2) Individual eye matching of the thalamic and total EPSCs (both ipsi and contra)*

*3) Inter-ocular difference of the total EPSC*

*4) Inter-ocular difference of the thalamic EPSC*

*5) Correlation of inter-ocular difference between total and thalamic EPSC*

*These properties should be presented in the exact same order also for young mice.*

We have followed this suggestion and rearranged this part of the Results. Now these properties are presented in the exact same order for young mice (subsection “Binocular matching of thalamic input precedes that of intracortical input”, second and third paragraphs).

*Furthermore, the intracortical EPSC and its individual eye matching with the thalamic EPSC as well as the inter-ocular difference of the intracortical EPSC should be presented with the rest of the EPSC properties in the adult first and then in the young animal.*

We tried to re-organize this part following the above suggestion, but found the flow awkward. So we decided to keep the results on intracortical EPSCs as a separate paragraph (last paragraph of Results). This way, the intracortical EPSC results can be more directly compared between young and adult mice, thereby highlighting one of the two developmental profiles we discovered in this study. It also avoids jumping back and forth between Figure 3 and Figure 4 in the text.

3) These are difficult experiments and the authors should acknowledge somewhere the limitations of their approach, ideally in the Discussion.

*A) Voltage clamp* in vivo *may be incomplete due to space clamp issues. This could even have larger effects on intracortical than thalamic input since the latter are likely to occur more proximally.*

We have added a new paragraph to acknowledge this issue in the Discussion (subsection “Earlier maturation of contralateral intracortical circuits”, third paragraph).

*B) The authors should also acknowledge the limitation of linear subtraction as a method for estimating the cortical contribution. The interaction between thalamic and cortical input is likely not linear.*

We have added a new paragraph to acknowledge this issue in the Discussion (subsection “Earlier maturation of contralateral intracortical circuits”, third paragraph).

*C) The discussion of possible early matching mechanisms in the Discussion is fine, but the statement "Amazingly, certain degrees of binocular correlation may indeed exist in mice even at the level of retinal waves (Ackman et al., 2012), possibly via direct retino-retinal connections (Muller and Hollander, 1988), retinopetal modulations (Gastinger et al., 2006), or light-driven activity in the intrinsic photosensitive retinal ganglion cells through the closed eye lids (Delwig et al., 2012; Renna et al., 2011)" implies that all such correlation must have occurred before eye opening. However, the experience between eye opening and recording may also contribute.*

We agree with this comment and added the following sentence before discussing experience-independent mechanisms. “It is possible that the visual experience between eye opening (P12-14) and our recording (P15-21) may contribute to this early thalamic binocular correspondence”.

*D) Finally, the following conclusion is not warranted, although it could be stated as a speculation: "feed-forward thalamocortical pathway plays an important role in initiating and guiding the functional rewiring and refinement of cortical circuits in visual system development." Correlation does not equal causation.*

We agree with this comment and have changed our writing to reflect this, from “…plays an important role…” to “…may play an important role…” (in the Abstract) and “the feed-forward thalamocortical pathway may play an important role in the development of visual circuits, possibly through initiating and guiding the functional rewiring and refinement of cortical circuits in visual cortex” (in the last paragraph of the Discussion).

*4) The scale factor was calculated as the maximal thalamic EPSC amplitude of the tuning curve divided by maximal total EPSC amplitude. Since preferred orientation often differed between thalamic EPSC and total EPSC, especially for young cells, the scale factor was often calculated from two responses evoked by different stimuli. To have a more accurate measurement of scale factor, the authors can select 4 o 5 best responses of the tuning curves for thalamic EPSC and total EPSC (evoked by the same stimuli), do some averaging and then calculate the ratio.*

We have changed the way of calculating scale factor following this suggestion. Instead of using the maximum amplitude or choosing 4-5 best responses, we now use the entire tuning curve. We averaged the responses across all directions for thalamic EPSC and total EPSC and then used their mean values to calculate the scale factor. The results are updated for both the figures (Figure 2—figure supplement 2) and the texts. The exact values were slightly different, but all conclusions held true.